# ACTIVATION REWARD MODELS FOR FEW-SHOT MODEL ALIGNMENT

## ABSTRACT

Aligning Large Language Models (LLMs) and Large Multimodal Models (LMMs) to human preferences is crucial to improving their real-world behaviour. A common approach is to use reward models to encode preferences, enabling alignment via reinforcement-learning post-training. However, traditional reward modeling is not easily adaptable to new preferences because it requires finetuning a separate reward model on large preference datasets. To address this, we introduce Activation Reward Models (Activation RMs)—the first mechanistic interpretability approach that steers LLM activations to better align with few-shot preference data without finetuning. Activation RMs is a novel steering method specifically designed for reward modeling by combining activation denoising and output token likelihood scoring to yield state-of-the-art performance, surpassing zero-shot, few-shot, and voting-based baselines on standard reward modeling benchmarks. Furthermore, we demonstrate the effectiveness of Activation RMs in mitigating reward hacking behaviors, showing that our approach is robust to noisy exemplars and spurious reward signals, highlighting its utility for safety-critical applications. Toward this end, we propose PreferenceHack, a novel few-shot benchmark that tests reward models on reward hacking in a paired preference format. We further show that Activation RM achieves state-of-the-art performance on this benchmark, surpassing even GPT-4o.

## 1 INTRODUCTION

Aligning Large Language Models (LLMs) [45, 58] and Large Multimodal Models (LMMs) [2, 32, 38, 61] with human preferences has become increasingly important in diverse applications such as question answering [41, 68, 73], summarization [54], and retrieval [70]. While traditional fine-tuning approaches effectively improve generative performance, they predominantly optimize more general next-token prediction objectives, which may not necessarily align with human intents on specific tasks. To address this problem, reward modeling and preference optimization have emerged as essential paradigms for post-training alignment to human preferences [4, 41]. However, traditional reward modeling requires large preference datasets and separate reward models for each new task or preference, limiting rapid adaptation to emerging safety threats or specific biases.

Recent approaches used LLMs as zero-shot reward models without finetuning [5, 30], including LLM-as-a-Judge [17] and token probability scoring methods [33, 72]. However, these generative reward models underperform specialized reward models and can be exploited through reward hacking [14, 60], even after extensive red-teaming [44, 47]. These challenges underscore the need for reward modeling approaches that can rapidly adapt using only few-shot examples [27] while maintaining robustness against exploitation.

To address these limitations, we propose **Activation Reward Models (Activation RMs)**—the first mechanistic interpretability [20, 24, 35] approach designed specifically for few-shot reward modeling. Our method is composed of three parts, each addressing a critical challenge in existing approaches. Given a particular reward modeling task, we first leverage few-shot examples to select attention heads well-suited for the preference objective. Our method's selection of specific heads enables more precise task alignment than other few-shot approaches like in-context learning [10]. Furthermore, human preferences lack the clear, verifiable metrics of standard tasks like VQA or captioning: preference labels are imperfect estimators of underlying human values and intentions. Thus, the

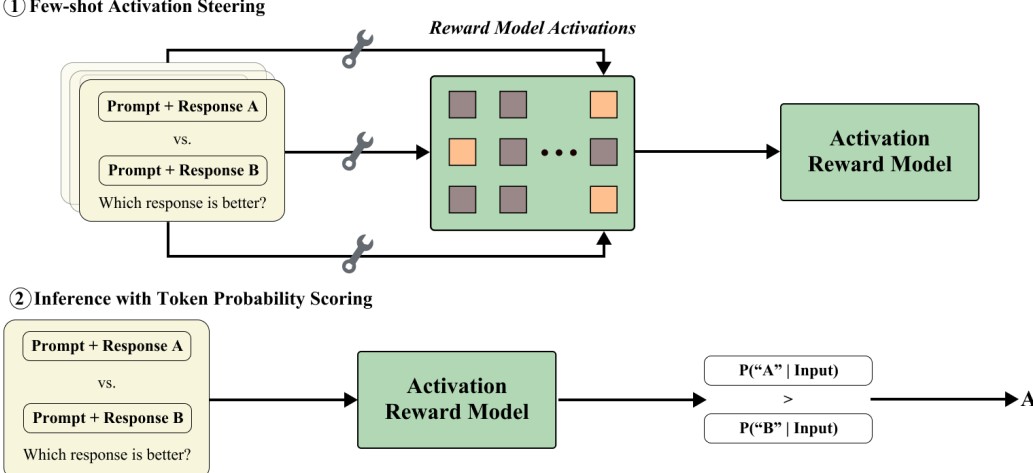

Figure 1: **Activation Reward Models.** The Activation RMs pipeline has two high-level steps. First, few-shot examples are used to steer specific attention heads within the model. Second, using this edited model, downstream inference for reward modeling is done via token probability scoring.

second component of our method employs a weighted variant of PCA to extract the underlying preference signal from few-shot activations by combining the top principal components weighted by their explained variance ratios. With the noise of any outlier labels having been filtered out, these activations are then replaced at the selected heads for steering. Third, we address the variability and hallucination caused by generative LLM-as-a-Judge approaches by leveraging generative token probability scoring rather than free-form generation. This approach offers a more concrete and consistently reliable reward signal than extracting a preference from natural text.

To rigorously evaluate reward model robustness, we introduce **PreferenceHack**, the first benchmark specifically designed to test reward hacking vulnerabilities through paired preference evaluation in a few-shot setting. Unlike existing benchmarks that focus on standard preference accuracy, PreferenceHack systematically probes models for exploitable biases such as length and format preferences, and evaluates the models' ability to mitigate such biases given few-shot exemplars.

Our contributions are as follows: (i) We introduce Activation RMs, a novel mechanistic interpretability framework that rapidly adapts to new preferences using only a handful of examples, outperforming existing few-shot reward modeling approaches on RewardBench and MultimodalRewardBench without any parameter updates; (ii) We present PreferenceHack, the first benchmark for evaluating reward hacking in paired preference formats; (iii) We demonstrate that our approach achieves state-of-the-art robustness against reward hacking, surpassing even GPT-4o while maintaining the flexibility to adapt to novel biases with just few-shot examples.

## 2 RELATED WORK

**Activation Steering and Task Vector Methods**. Recent advances in mechanistic interpretability and activation-based control methods have revealed how model behavior can be precisely manipulated through internal representations. Early research in neural network interpretability [8, 9, 74] established frameworks for understanding how individual neurons encode semantic concepts across network layers, while activation steering methods [42, 55, 59] demonstrated that behavior modification could be achieved without parameter updates. Building on these foundations, the discovery of specialized components (e.g., induction heads [37, 66], task-specific neurons [21]) led to task vector abstractions for capturing and manipulating computational patterns within models [19, 57].

Following these insights, researchers extended activation-based approaches to multimodal settings through visual task vectors [22], multimodal task vectors [25], and sparse attention vectors [35]. These methods build on the observation that task-relevant information is often concentrated in specific attention heads or activation subspaces, enabling efficient context summarization and few-

shot learning under limited context. Parallel work in understanding multimodal representations [49] and language-guided visual editing [16] has further highlighted how multimodal models structure and manipulate cross-modal concepts via localized activations. While prior methods have shown success, our work is the first to apply few-shot activation steering to reward modeling, integrating it with token probability scoring for fast adaptation to new tasks without parameter updates or added context.

**Reward Modeling**. Early work showed reinforcement learning could leverage human feedback instead of hand-crafted reward functions [13, 53, 75]. The standard RLHF pipeline trains a reward model on human preference data before optimizing a policy against this reward [3, 40], typically using PPO [48]. More recent approaches simplify this process: Direct Preference Optimization (DPO) [46] derives the optimal policy in closed-form, while ranked response methods [11, 69] and guided optimization [50] offer alternatives to full RL. Reward models traditionally share the LLM architecture with an added scalar output [3, 40], though newer approaches include LLM-as-judge prompting [17] and Generative Verifiers [71] that produce reasoning steps before judgment. Research has also shown that AI feedback can replace human feedback with comparable results but greater scalability [6, 29]. Benchmarks like RewardBench [28] and its multilingual [18], retrieval-based [26], and adversarial [34, 63] variants have emerged to standardize reward model evaluation, with Multimodal RewardBench [65] extending this to vision-language models. Few-shot preference learning approaches include meta-learning-based Few-Shot Preference Optimization (FSPO) [51], In-Context Preference Learning (ICPL) [67], feature-based methods [7], and Rule-Based Rewards [36] that encode behaviors in written rules. In contrast to these approaches that require fine-tuning, prompting, or complex RL, our Activation RMs leverage activation steering to construct accurate reward models from minimal examples with no additional training, representing the first application of activation steering to the reward modeling problem.

## 3   ACTIVATION REWARD MODELS

While traditional reward modeling effectively aligns LLMs and LMMs to human preferences, it fundamentally lacks adaptability due to its dependence on large labeled datasets and extensive training. We present Activation Reward Models (Activation RMs), a framework that enables precise reward modeling with minimal examples and no additional training through three targeted components: activation steering for task specification, weighted PCA denoising for robust preference extraction, and generative scoring for reliable evaluation. Figure 1 illustrates our approach.

### 3.1   PROBLEM SETUP

In reward modeling, given responses $r$ to a prompt $p$, a reward model $R$ evaluates alignment with human preferences—either as a scalar score for a single response or as a preference between multiple responses. Traditional approaches require extensive preference datasets and separate model training. In contrast, few-shot reward modeling constructs accurate reward signals using only a small set of examples $\{(p_i, r_i, y_i)\}_{i=1}^n$ where $y_i$ indicates the preference outcome (whether a response meets criteria or which response is preferred). Activation RMs leverage these few examples to adapt to new preference specifications without parameter updates.

### 3.2   ATTENTION HEAD SELECTION AND ACTIVATION EXTRACTION

Unlike in-context learning which relies on surface-level patterns, we directly modify the model's internal representations to encode preference criteria. We begin by identifying which attention heads best capture preference evaluation and extracting their activations.

A transformer with $L$ layers and $H$ attention heads processes inputs through multi-head self-attention where in each layer $l \in \{1, \ldots, L\}$ and head $m \in \{1, \ldots, H\}$, the attention mechanism computes:

$$\mathbf{h}_l^m(x_i) = \text{softmax}\left(\frac{QK^T}{\sqrt{d_m}}\right) V$$

where $Q, K, V$ are the query, key, and value matrices, and $d_m = d/H$ is the dimensionality per head. We denote $\mathbf{h}_l^m(x_i)$ as the attention vector for head $m$ in layer $l$ at position $i$.

For each few-shot triple $(p_i, r_i, y_i)$, we wrap the task in a pairwise template. When the model runs this template, we read last token activations $z_{l,m,j}$ at head $(l, m)$ for criterion $j$. To choose the heads that

encode the criterion, we optimize a Bernoulli over head indicators with REINFORCE [62]: sample a binary mask over heads, evaluate accuracy on a validation split, and update inclusion probabilities toward higher accuracy masks, yielding an optimized select set $\lambda_j^{\text{ARM}}$.

## 3.3 WEIGHTED PCA DENOISING FOR ROBUST PREFERENCE EXTRACTION

Human preference labels contain inherent noise from annotator disagreements and inconsistent criteria application. Rather than using simple averaging which treats all activation dimensions equally, we apply weighted PCA to extract the core preference signal.

We run PCA on the activation vectors $z_{l,m,j}$ from the selected heads over all few-shot examples, yielding components $v_1, \ldots, v_k$ with variance weights $w_1, \ldots, w_k$ that quantify how much preference signal each captures.

To denoise the activations, we compute a weighted average of the top-$k$ principal components: $\mu_j^{\text{ARM}} = \sum_{i=1}^{k} w_i \cdot v_i$ where $w_i$ is the explained variance ratio of the $i$-th principal component, normalized across the top-$k$ components. This weighted combination prioritizes the dimensions that capture the most variance in the preference signal while filtering out noise from less informative dimensions, making our method robust to label inconsistencies and annotation errors.

## 3.4 GENERATIVE SCORING FOR FORMALIZED EVALUATION

Instead of free-form generation which is prone to randomness and hallucinations we score via token probabilities. For a new response $r$ to prompt $p$, we inject denoised activations $\mu_j^{\text{ARM}}$ at the selected attention head locations $\lambda_j^{\text{ARM}}$ and query the model:

$$s(r \mid p) = P_F(\text{``Yes''} \mid \text{``Does this response meet the specified criteria?''}, \lambda_j^{\text{ARM}}, \mu_j^{\text{ARM}})$$

The reward score is the probability of generating "Yes", providing a calibrated scalar signal that directly leverages the model's understanding without additional training. This approach eliminates the inconsistencies of language-based judgments while maintaining interpretability.

## 3.5 IMPLEMENTATION AND APPLICATIONS

Activation RMs naturally extend to multimodal inputs by incorporating visual information into the prompt structure, enabling consistent preference evaluation across modalities. The framework's flexibility supports diverse applications: serving as a general evaluator by adapting evaluation criteria, enabling best-of-N sampling through response ranking, or providing scalar rewards for reinforcement learning-based preference optimization. Importantly, all adaptation occurs through the few-shot examples alone—no architectural changes or parameter updates are required, making Activation RMs immediately deployable for new preference specifications.

# 4 PREFERENCEHACK: A FEW-SHOT REWARD HACKING BENCHMARK

Reward hacking—where certain model biases exploit confounding factors in reward functions rather than satisfying the intended objectives—remains a significant challenge for alignment. To evaluate the robustness of reward models against such exploitation, we introduce PreferenceHack, a novel evaluation benchmark specifically designed to assess reward models' susceptibility to common bias-based reward hacking behaviors.

## 4.1 BENCHMARK DESIGN

To the best of our best knowledge, PreferenceHack is the first benchmark that evaluates reward hacking *in a few-shot setting with a paired preference format*, allowing direct assessment of reward models' vulnerabilities to known biases.

The benchmark consists of six distinct splits across language and multimodal domains, with each split containing 80 few-shot training examples and 920 evaluation examples. This structure allows

robust evaluation of reward models across diverse bias conditions with strong statistical power. More details about our dataset and its construction are included in Sec C.2 of the Supp.

## 4.2 DATASET CONSTRUCTION

### 4.2.1 LANGUAGE SPLITS

For the language-based splits, we built upon findings from the "Helping or Herding?" study [15], which documented exploitable biases in language models. We used high-quality ground truth answers from the original dataset and generated preference pairs by systematically injecting three well-known biases into the incorrect samples: (i) **Length Bias**: Models often assign higher scores to longer responses regardless of content quality. We generated longer alternatives to the incorrect responses while preserving their factual inaccuracies; (ii) **Format Bias**: Structured formats like numbered lists often receive higher scores despite potential content issues. We reformatted incorrect responses into structured formats to exploit this bias; and (iii) **Positivity Bias**: Responses containing positive attitudes tend to score higher. We injected positive tone into incorrect responses to trigger this bias.

To ensure consistency in generating the non-preferred responses, we used GPT-4o-mini to inject the bias being evaluated into the incorrect response.

### 4.2.2 MULTIMODAL SPLITS

For multimodal evaluation, we created three splits using image-prompt pairs from SUGAR-CREPE [23], a challenging compositional image-text retrieval dataset. Each pair in the multimodal split of PreferenceHack contains an image with a correct and incorrect prompt description. Similar to our language splits, we used GPT-4o-mini to inject model biases into the incorrect descriptions while preserving their factual errors. This approach creates a test bed for assessing multimodal reward hacking vulnerabilities.

## 4.3 EVALUATION PROTOCOL

PreferenceHack employs a few-shot evaluation protocol where reward models are exposed to a small set of examples (80 per split) before being evaluated on the larger test set (920 examples per split). This format specifically tests the ability of reward models to quickly adapt to model biases given few-shot examples. We show some examples of our benchmark in Figure 2.

For each preference pair, a reward model is considered successful if it assigns a higher reward score to the correct response compared to the biased alternative. This simple evaluation metric directly measures a reward model's susceptibility to common exploitation patterns.

## 5 EVALUATION

We evaluate Activation RMs across a diverse set of benchmarks to assess their effectiveness as few-shot reward models and their ability to mitigate reward hacking. We apply our approach to two state-of-the-art Large Multimodal Models: LLaVA-OneVision-7B and Qwen2.5-VL-7B. Our experiments focus on comparing Activation RMs against existing few-shot approaches in standard reward modeling tasks, evaluating robustness against reward hacking, and assessing performance on multimodal retrieval tasks.

## 5.1 IMPLEMENTATION DETAILS

We implemented Activation RMs using PyTorch [43]. We used the official implementations of LLaVA-OneVision-7B [31] and Qwen2.5-VL-7B [2] as base models. All experiments were conducted on a single NVIDIA A100 GPU with 80GB memory. For the activation steering procedure directly edit the output of each attention head before the projection layer.

For each experiment, we used a consistent few-shot setting with $n \leq 130$ examples for training our Activation RMs unless otherwise specified. The activation extraction process involves collecting attention head activations from the last token of the input prompt. For attention head selection, we

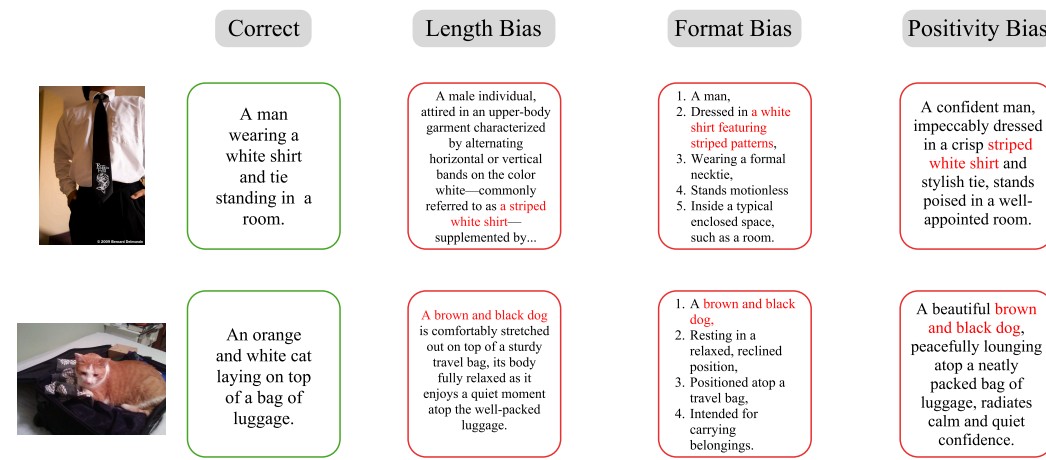

Figure 2: **PreferenceHack Examples.** We show samples based on two images of our PreferenceHack benchmark. Each sample would consist of a ground truth response paired with a biased incorrect response. The reward model is tasked with preferring the correct description over the biased one.

Table 1: **Evaluation of Activation RM on RewardBench and Multimodal Reward Benchmarks.** We perform a thorough evaluation of Activation RMs and baselines across multiple splits in language-only and multimodal settings. We present GPT-4o as a reference closed-source result.

| Method / Model | Language-Only (RewardBench) | | | | | | Multimodal (Multimodal RewardBench) | | | | | | | | |
|---|---|---|---|---|---|---|---|---|---|---|---|---|---|---|---|
| | Safety (%) | Chat (%) | Chat Hard (%) | Reasoning (%) | Overall (%) | Macro Avg. (%) | Correct. (%) | Pref. (%) | Knowl. (%) | Math (%) | Coding (%) | Safety (%) | VQA (%) | Overall (%) | Macro Avg. (%) |
| GPT-4o | 85.74 | 94.74 | 73.01 | 90.93 | 87.63 | 86.10 | 50.91 | 48.09 | 60.20 | 59.11 | 54.42 | 85.19 | 47.48 | 55.43 | 57.92 |
| *LLaVA-OneVision-7B* | | | | | | | | | | | | | | | |
| ZS LLM-as-a-Judge | 68.85 | 82.89 | 40.49 | 52.81 | 57.93 | 61.26 | 53.54 | 51.81 | 55.28 | 53.14 | 57.88 | 4.90 | 49.82 | 48.04 | 46.62 |
| 8-shot LLM-as-a-Judge | 58.69 | 43.42 | 45.09 | 49.65 | 50.71 | 49.21 | 57.61 | 59.16 | 55.80 | 58.07 | 50.22 | 38.10 | 46.07 | 51.57 | 52.15 |
| ZS Generative Scoring | 49.51 | 55.26 | 50.61 | 47.43 | 49.09 | 50.70 | 48.88 | 49.05 | 48.00 | 52.60 | 50.00 | 49.21 | 50.84 | 49.88 | 49.80 |
| 3-sample voting | 67.21 | 84.21 | 40.80 | 52.73 | 57.65 | 61.24 | 56.19 | 54.39 | 56.20 | 53.91 | 56.86 | 5.29 | 49.81 | 48.93 | 47.52 |
| SAV | 69.40 | 85.70 | 45.60 | 65.20 | 64.50 | 66.47 | 55.50 | 53.20 | 54.80 | 53.50 | 56.90 | 40.30 | 49.50 | 51.80 | 52.00 |
| Activation RM | 70.98 | 88.60 | 50.31 | 69.02 | 68.84 | 69.73 | 49.90 | 48.56 | 54.91 | 52.90 | 50.62 | 81.62 | 49.00 | 53.75 | 55.36 |
| *Qwen2.5-VL-7B* | | | | | | | | | | | | | | | |
| ZS LLM-as-a-Judge | 75.90 | 88.16 | 58.59 | 70.64 | 71.97 | 73.32 | 65.92 | 64.89 | 59.20 | 57.03 | 57.30 | 79.63 | 74.95 | 66.88 | 65.56 |
| 8-shot LLM-as-a-Judge | 80.00 | 87.72 | 61.35 | 73.02 | 74.56 | 75.52 | 64.30 | 64.89 | 60.60 | 58.07 | 54.87 | 76.19 | 73.74 | 65.98 | 64.66 |
| ZS Generative Scoring | 50.00 | 46.05 | 52.45 | 50.19 | 50.06 | 49.67 | 61.66 | 62.98 | 48.20 | 53.12 | 53.76 | 49.21 | 62.24 | 57.20 | 55.88 |
| 3-sample voting | 77.05 | 89.91 | 57.67 | 69.18 | 71.52 | 73.45 | 66.53 | 64.70 | 59.00 | 56.77 | 59.29 | 80.16 | 74.58 | 67.06 | 65.86 |
| SAV | 76.50 | 90.20 | 56.80 | 74.30 | 74.50 | 74.45 | 64.50 | 62.50 | 58.70 | 56.53 | 54.77 | 100.00 | 72.00 | 67.50 | 67.00 |
| Activation RM | 78.03 | 94.74 | 57.06 | 78.86 | **77.24** | **77.17** | 63.29 | 65.84 | 56.40 | 59.64 | 60.18 | 98.15 | 76.82 | **69.27** | **68.62** |

use 600 optimization steps with the REINFORCE algorithm [62]. Additional implementation details and hyperparameters can be found in the Appendix.

## 5.2 DATASETS

We evaluate Activation RMs on three paired preference datasets where models must identify the preferred response between two candidates: (i) **RewardBench** [28] and **MultimodalRewardBench** [64] are comprehensive reward modeling benchmarks that evaluate out-of-the-box pretrained LLMs and LMMs on a variety of different language-only and multimodal tasks; in both benchmarks, given a prompt, the model must choose between a preferred and non-preferred response; (ii) **Preference-Hack** evaluates reward models' susceptibility to reward hacking with seven splits (80 training, 920 evaluation examples each) across language and multimodal domains. It systematically injects biases (length, format, numerical, and orientation) to assess how quickly reward models can identify and mitigate exploitation patterns with minimal examples. More details are in Section C.2 of the Supp.

## 5.3 BASELINES

We compare Activation RMs against several established reward modeling approaches: **LLM-as-a-Judge** prompts the model to directly output a preferred response given a pair in either zero-shot or few-shot (8 examples) settings; **Generative Verifier** [33, 72] derives preferences by comparing the

Table 2: **Evaluation of Activation RM on PreferenceHack Benchmark.** We thoroughly evaluate Activation RMs and baselines on our novel few-shot reward hacking benchmark: PreferenceHack. We present GPT-4o as a reference closed-source result.

| Method / Model | Language-Only Splits | | | Multimodal Splits | | |
|---|---|---|---|---|---|---|
| | Length (%) | Format (%) | Positivity (%) | Image+Length (%) | Image+Format (%) | Image+Positivity (%) |
| GPT-4o | 3.91 | 48.04 | 92.39 | 22.35 | 55.78 | 87.65 |
| *LLaVA-OneVision-7B* | | | | | | |
| ZS LLM-as-a-Judge | 14.46 | 44.89 | 59.24 | 28.30 | 51.20 | 54.75 |
| 8-shot LLM-as-a-Judge | 23.15 | 37.50 | 57.17 | 38.45 | 45.65 | 52.30 |
| ZS Generative Scoring | 45.54 | 47.17 | 76.96 | 57.80 | 54.25 | 71.40 |
| 3-sample voting | 15.43 | 43.26 | 59.67 | 30.85 | 49.75 | 55.10 |
| SAV | 45.80 | 75.30 | 86.45 | 60.25 | 78.40 | 80.65 |
| Activation RM | 49.24 | **79.89** | 90.11 | 65.70 | **83.45** | 85.25 |
| *Qwen2.5-VL-7B* | | | | | | |
| ZS LLM-as-a-Judge | 1.41 | 41.63 | 88.70 | 18.75 | 48.30 | 82.15 |
| 8-shot LLM-as-a-Judge | 8.70 | 47.39 | 87.28 | 25.40 | 53.85 | 80.60 |
| ZS Generative Scoring | 17.72 | 50.65 | 93.59 | 35.20 | 58.40 | 88.25 |
| 3-sample voting | 1.41 | 41.85 | 88.70 | 19.30 | 48.75 | 82.50 |
| SAV | 73.50 | 65.75 | 93.80 | 78.65 | 70.35 | 88.90 |
| Activation RM | **78.37** | 68.26 | **96.74** | **84.25** | 75.50 | **91.80** |

probability of a "Yes" token when asked if responses meet specified criteria; **3-Sample Voting** - A natural language reward modeling approach that implements self-consistency through a chain-of-thought methodology. The model generates three independent evaluations for each response, and the final preference is determined by majority voting across these samples;**Sparse Attention Vectors (SAVs)** [35] - A method that leverages few-shot examples to extract features from the attention heads of a model for classification, enabling another comparable SOTA form of few-shot reward modeling.

## 6 RESULTS

We perform a thorough evaluation of our Activation Reward Model on multiple benchmarks and compare to a variety of baselines. We first present the results of our few-shot method on general reward model benchmarks which for each group used a maximum of 130 examples for activation steering. Following this, we focus on the application of our method to the domain of safety and reward hacking on our PreferenceHack benchmark which we used 80 examples per group for steering. Finally, we perform several ablations and additional experiments to probe important characteristics of our approach.

### 6.1 REWARD BENCHMARK RESULTS

We perform evaluation on two comprehensive reward model benchmarks (Language-Only) Reward-Bench [28] and Multmodal RewardBench [64] as shown in Table 1. On average across a variety of splits, our Activation Reward Model outperforms all zero-shot and few-shot open-source baselines on both language-only and multimodal benchmarks, suggesting the effectiveness of our approach. Furthermore, our approach closes the gap with a strong closed-source baseline such as GPT-4o. This is especially important as GPT-4 and other closed source models are often used as a reward models or judges of open-source models' outputs. However, a clear advantage of our approach is the interpretability of using few-shot examples of a task to specify a reward signal. Thus, our approach is both a more aligned and interpretable reward score for model alignment. Interestingly, our results show that few-shot, generative verification, and voting baselines struggle to outperform zero-shot LLM-as-a-Judge, suggesting that reward modeling is a challenging domain for these common SOTA methods. This further highlights the effectiveness of Activation RM. Additional results are provided in Section A of our Supp.

Finally, we highlight our method's consistent gains on safety tasks, which we attribute to a property we refer to as *taskness*. To clarify, safety tasks are more well-defined and thus better captured through few-shot examples, which Activation RMs use to guide reward model activations. In contrast, domains like chat, reasoning, or math are broader and less specific. For example, a few safety

Table 3: **Ablations.** We conduct ablations on Activation RMs using Qwen2.5-VL on RewardBench.

| Ablation Method | Safety (%) | Chat (%) | Chat Hard (%) | Reasoning (%) | Overall (%) | Macro Avg. (%) |
|---|---|---|---|---|---|---|
| ZS LLM-as-a-Judge | 75.90 | 88.16 | 58.59 | 70.64 | 71.97 | 73.32 |
| CoT baseline | 73.93 | 88.60 | 51.23 | 69.95 | 70.18 | 70.93 |
| CoT + Voting | 74.59 | 89.47 | 52.15 | 70.25 | 70.71 | 71.62 |
| LoRA Finetuning | 77.50 | 92.41 | 59.44 | 72.40 | 73.56 | 73.51 |
| Mean Activation Addition | 65.82 | 81.37 | 42.15 | 61.28 | 62.47 | 62.66 |
| Top PCA Vector Replacement | 76.24 | 91.58 | 54.91 | 75.93 | 74.73 | 74.67 |
| Mean Activation Difference | 76.51 | 92.85 | 55.32 | 77.24 | 75.48 | 75.48 |
| **ActivationRM** | 78.03 | 94.74 | 57.06 | 78.86 | 77.24 | 77.17 |

examples clearly define safe vs. unsafe responses, while a few math examples are less informative due to the diversity of sub-tasks (e.g., geometry, number theory, complex analysis). Thus, we posit that our approach and few-shot reward modeling methods more generally may be more successful when the application is to a *well-specified* or more *fine-grained* task.

## 6.2 PreferenceHack Results

To evaluate the effectiveness on a critical safety-like task, we apply our method to our new benchmark PreferenceHack as shown in Table 2. When evaluated on multiple different reward hacking biases in both language-only and multimodal settings, we find our method significantly outperforms all baselines in protecting against common reward hacks, even outperforming GPT-4o on most splits. Reward hacking is a task that quickly changes as new methods are found to exploit model biases. Hence, our approach is perfectly suited for adapting a reward model to be robust to a new attack given just a few examples.

## 6.3 Ablation Studies

We explore different properties and capabilities of our framework via a careful ablation study in Table 3 using Qwen2.5-VL-7B evaluated on RewardBench.

**Effect of CoT on Activation RMs**. We are also interested in how the common approach of generating a CoT reasoning chain before outputting a preference impacts Acitvation RM. To do this, examples are formulated using the prompt, responses, and a chain-of-thought reasoning chain. Inference is performed in two steps. First, a CoT reasoning chain is generated given the prompt and two responses. Then, the final preference is outputted conditioned on the prompt, responses, *and* CoT reasoning chain. We find interestingly that CoT reasoning in this manner has little effect on our results, suggesting a future area of exploration for Activation RM.

**Activation RM Comparable w/ LoRA Finetuning**. We are also motivated to compare our framework with the common approach of finetuning an LLM/LMM explicitly as a reward model. We apply rank 16 LoRA finetuning for 3 epochs using 130 examples. Interestingly, we find that Activation RM yields similar performance as finetuning a model for reward modeling. This demonstrates that our method is both effective as a reward model and sample-efficient, requiring no weight updates to the generative model solely for reward modeling.

## 6.4 Additional Results

**Superior Robustness to Label Noise**. Figure 3 reveals ActivationRM's resilience to label corruption across all PreferenceHack splits. While LoRA fine-tuning exhibits catastrophic degradation under noise dropping by over 50% in some cases ActivationRM maintains stable performance even with 30% label corruption. This robustness stems from our design choices: weighted PCA filters out noisy variations in the calibration data, while output token likelihood scoring provides a more stable signal than methods that directly optimize on potentially mislabeled examples. The activation-based approach effectively reduces multiple sources of variability that plague traditional fine-tuning and prompting methods under noisy conditions.

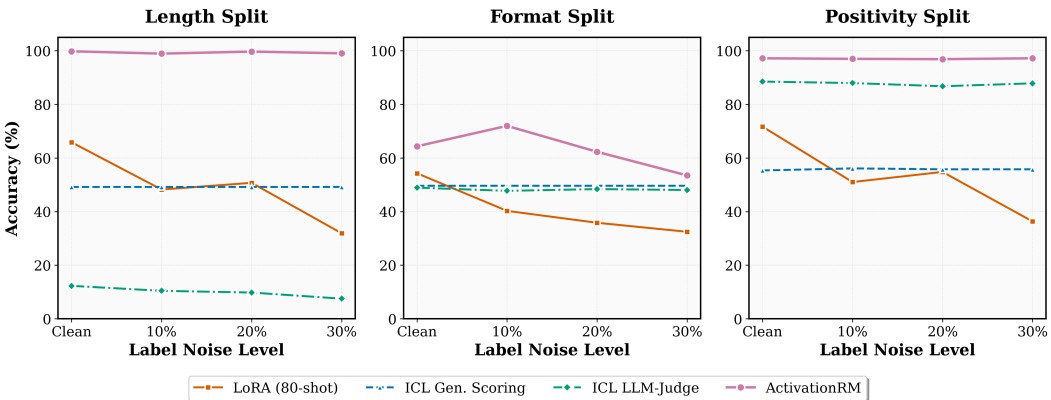

Figure 3: Robustness to label noise on PreferenceHack language splits. We evaluate ActivationRM against three baselines (LoRA fine-tuning, ICL Generative Scoring, and ICL LLM-as-a-Judge) across increasing levels of label noise (0% to 30%). Results are shown for Length, Format, and Positivity preference splits.

# 7 CONCLUSION

We introduce Activation Reward Models, the first mechanistic interpretability approach designed for few-shot reward modeling. By combining activation steering for precise task specification, weighted PCA denoising for robust preference extraction, and generative scoring for reliable evaluation, our method achieves state-of-the-art performance without any parameter updates. Our comprehensive evaluation demonstrates that Activation RMs consistently outperform existing few-shot approaches on both language-only and multimodal benchmarks, surpassing even GPT-4o on our novel PreferenceHack benchmark while providing greater interpretability through explicit few-shot examples.

Without extensive data collection or model retraining, the framework's flexibility enables fast deployment across diverse applications—from general evaluation tasks to best-of-N sampling and reinforcement learning—with adaptation occurring solely through few-shot examples. Our ablations suggest that performance can scale with more examples while maintaining few-shot practicality, and that our approach achieves comparable results to LoRA fine-tuning without requiring any weight updates. By enabling models to adapt to evolving preferences and emerging safety threats as shown by strong peformance on our novel PreferenceHack benchmark, Activation RMs provide a practical path toward more adaptive and robust AI alignment.

# 8 LIMITATIONS

Activation Reward Models represent a significant advancement in few-shot reward modeling, but several limitations should be acknowledged. First, our approach requires access to a model's internal architecture to extract and manipulate attention head activations, making it inapplicable to closed-source models like GPT-4o [39] and Claude [1]. Second, while Activation RM performs well on our benchmarks, the method's effectiveness may diminish for tasks that are less well-specified or require understanding of a broad range of criteria that cannot be captured in a few examples, such as mathematics. Finally, the current implementation focuses on single-turn interactions, and extending the approach to multi-turn dialogues or longer contexts may require additional research on how activation steering propagates across extended sequences. These limitations highlight opportunities for future work in developing more robust few-shot reward modeling techniques that can operate with more limited model access or handle more complex evaluation scenarios. Finally, we do not anticipate specific negative impacts, but as with any machine learning method, we recommend exercising caution in deployment.

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
