# OpenReview forum: "Activation Reward Models for Few-Shot Model Alignment"
_ICLR.cc/2026/Conference — ICLR 2026 Conference Withdrawn Submission_

### Official Review · Reviewer_GNqj · 2025-10-19

**Soundness:** 2
**Presentation:** 1
**Contribution:** 2
**Rating:** 2
**Confidence:** 3

**Summary:**

The paper proposes Activation Reward Models (ARM), a method for reward modelling via identifying and modifying specific internal activations based on fews-shot preference data without fine-tuning. The approach uses REINFORCE to select attention heads whose activations are modified (“injected”) at test time. The method is evaluated on three bias-correction tasks (length, format, and positivity).

While the idea of directly manipulating activations for alignment is intriguing, the paper suffers from significant clarity and methodological issues. Key definitions and implementation details are missing, making it difficult to reproduce or interpret results. The evaluation is limited and does not convincingly demonstrate the method’s claimed advantages.

**Strengths:**

- Interesting direction: The notion of activation-level alignment—intervening on internal representations instead of fine-tuning—is conceptually novel and could inspire future research.

**Weaknesses:**

1. **Poorly explained methodology:** The core mechanism of ARM is underspecified. It is unclear what the criterion ($j$) refers to, what ( \lambda_j^{\text{ARM}} ) represents (a set of locations $(l, m)$?), or how activations are “injected” at selected head locations.

2. **Unclear use of REINFORCE:** The paper claims to use REINFORCE for head selection but does not define what constitutes the “reward” signal for this optimization or how the stochastic policy over heads is parameterized.

3. **Evaluation setup:**
   * The evaluation design risks confounding effects — PreferenceHack constructs incorrect responses by injecting a specific bias (length/format/positivity) into incorrect samples using a generator (GPT-4o-mini) while preserving factual inaccuracies (Sec. 4.2). This construction introduces a deterministic correlation between correctness and the opposite of the injected bias (e.g., if incorrect = longer, then correct = shorter). The authors do not report experiments where the mapping is flipped at test time (e.g., in-context examples indicate long→correct but test pairs use long→incorrect). Therefore their method might be learning the superficial mapping between bias and correctness in the few-shot exemplars rather than the underlying correctness signal. The paper does not show a counterbalanced test or out-of-distribution swap that would resolve this concern.
   * The experiments test only three synthetic bias-correction tasks (length, format, positivity), which are simplistic and may not reflect real alignment challenges.
   * The claim of “few-shot” learning (80 examples) may be misleading; few-shot typically refers to only a handful of training examples (typically < 20).


4. **Lack of confidence intervals:** The reported results show means only, without confidence intervals, variance estimates, or statistical significance testing, despite small test sets (~920 per split).

5. **Unexplained bold formatting in tables:**  It is not explained what criterion is used to highlight in bold the results presented in Table 2.

6. **Unclear generalization:** It is unclear how the method would handle different evaluation settings. It solely focuses on the aspect of correctness in the presence of length, format, and positivity biases. It would be beneficial to see how the method can be applied for personalized alignment scenarios where the few-shot examples demonstrate a preference towards certain values that vary across different personas? (e.g. the setup of [1, 2, 3]).

**Questions:**

1. What precisely is an optimization criterion indexed by ($j$)?
2. What are the "specified criteria" in the evaluation prompt (section 3.4)?
2. What does ($\lambda_j^{\text{ARM}}$) denote precisely? How is the size of this set decided?
3. How are activations “injected” at selected locations? Does this mean additive perturbation or replacement?
4. How precisely is REINFORCE used — what is the policy, what is the reward, and what are the sampled actions?
5. How does performance scale with the number of training examples? Could the authors provide results for $n < 80$? E.g. $n=3, 5, 10,  20$ which is standard in the few-shot learning literature.
6. How robust is ARM to spurious correlations (e.g., if shorter responses are correlated with “correct” labels in training but not at test time)?
7. Could this approach handle personalized alignment tasks beyond the axis of correctness? (e.g. preference for certain moral values, or trade-offs in helpfulness vs. harmlessness as in [1, 2, 3]).

---

References:

[1] https://arxiv.org/abs/2408.10075 Personalizing Reinforcement Learning from Human Feedback with Variational Preference Learning

[2] https://arxiv.org/abs/2412.13998 Few-shot Steerable Alignment: Adapting Rewards and LLM Policies with Neural Processes

[3] https://arxiv.org/pdf/2502.19312 FSPO: Few-Shot Preference Optimization of
Synthetic Preference Data in LLMs Elicits Effective Personalization to Real Users

---

### Official Review · Reviewer_YDfE · 2025-10-21

**Soundness:** 1
**Presentation:** 2
**Contribution:** 1
**Rating:** 2
**Confidence:** 3

**Summary:**

This paper proposes activation reward models, which are claimed to be a novel steering method for addressing the problem of adapting to new preferences without requiring the retraining of a reward model from scratch. Therefore, the authors focus on the issue of few-shot reward learning. Moreover, the authors introduce a new benchmark, titled PreferenceHack, which is used to evaluate how reliable reward models are against reward-hacking (e.g. length, format, and positivity)

**Strengths:**

The paper addresses an interesting problem, as few-shot reward learning is a viable approach for personalisation in LLMs.
Moreover, I think the constructed PreferenceHack benchmark can help investigate reward hacking of positivity, length, and format of the preferences.

**Weaknesses:**

While the paper addresses an interesting problem, and I appreciate the authors' proposed method, I believe that this paper has some significant points that need to be improved upon, or at least addressed, by the authors:

- **Missing related work**: There are some key missing related works in the realm of few-shot preference learning. While I understand that most of these require retraining the reward model, it would be important to highlight these and compare the results [1, 2, 3]. Moreover, AlpacaEval 2.0 is an LLM-as-a-judge framework that considers length-reward hacking; thus, it is worth mentioning.

- **No statistical significance**: All the reported results do not show any confidence intervals, standard errors, or standard deviation. Therefore, it is challenging to determine the actual significance of the results. This is particularly concerning when looking at Table 5 in the appendix, where the different numbers of few-shots are tested on the models, and there seems to be no visible improvement when using more samples (2 shots MAv: 69.73%, 8 shots MAv: 68.29). In my opinion, there should be a clear performance improvement as the number of shots used in training increases. Moreover, the fact that even with only two shots the reward model can classify with ~70% accuracy means to me that the reward model was already somehow trained on this data.

- **Lack of method clarity**: The main paper lacks clarity in explaining the method, especially in paragraph 3.2. I am not entirely sure how the REINFORCE algorithm is used on the masks. I could find some more details in the appendix, but I think too many details are omitted in the main manuscript.

- **Lack of clarity when explaining the results**: I assume the values reported in Tables 1 and 2 are the accuracies of the reward model on the test set? The metric was never really explained, to the best of my knowledge.

**Misc**:
 - I would not use the "mechanistic interpretability" claim here, as the focus of your paper is not on interpretability
 - the maths in your background section should in my opinion be consistent with related works in preference learning having a dataset $\\{(x_i, y_{w, i}, y_{l,i})\\}_{i=0}^n$, where $y_w$ and $y_l$ are the winning and losing responses to the prompt $x$.
 - To me, the results in Figure 3 for "length" and "positivity" look suspiciously high and close to 100%, despite label noise. I would expect to see some effect as the noise level increases
- In my opinion, 80 samples is not really few-shot learning anymore, as that's the size e.g. some medical datasets have




[1] Poddar, S., Wan, Y., Ivison, H., Gupta, A., and Jaques, N. Personalizing Reinforcement Learning from Human Feedback with Variational Preference Learning, August 2024. URL http://arxiv.org/abs/2408.

[2] Kobalczyk, K., Fanconi, C., Sun, H., and van der Schaar, M. Few-shot Steerable Alignment: Adapting Rewards and LLM Policies with Neural Processes, December 2024. URL http://arxiv.org/abs/2408.

[3] Zhao, S., Dang, J., and Grover, A. Group preference optimization: Few-shot alignment of large language models, 2023

**Questions:**

I have the following questions:
- Are the results statistically significant?
- How do trained preference learning algorithms perform on the PreferenceHack dataset (i.e. what is our upper bound?)

---

### Official Review · Reviewer_L9GQ · 2025-10-24

**Soundness:** 2
**Presentation:** 2
**Contribution:** 2
**Rating:** 2
**Confidence:** 4

**Summary:**

This paper proposes Activation Reward Models for few-shot alignment. Specifically, it includes activation denoising and output token probability scoring to reward modelling. The proposed methods can align with few-shot preference data without fine-tuning. Additionally, this paper introduces a new benchmark, called PreferenceHack, to evaluate the reward models on reward hacking problem. Experiments demonstrate that the proposed methods outperform baselines on the proposed benchmark.

**Strengths:**

- It is very interesting to consider activation steering for reward modelling.
- Experiments show the effectiveness of the proposed method.
- The example presentation help me understand the proposed benchmark.

**Weaknesses:**

- The paper is not well written, which is hard to follow.
- It seems that the proposed method is a little bit overcomplex and not intuitive, including REINFORCE-based head selection, weighted PCA and token probability scoring, which is a complex pipeline containing multiple steps.
- It is unclear which step plays a more important role in your method.
- The baselines do not include baselines of the implicit reward modelling, such as DPO, IPO or SimPO.
- It seems that the performance on the RewardBench does not consistently outperform  the baselines, such as on "Safety" and "ChatHard".
- It seems that the reference is not cited in the ICLR style.

**Questions:**

- How do existing LLMs perform on your proposed benchmark, including open source and close source LLMs?
- Given Table 1, your proposed method underperform baselines on many situations, could you provide some explanation for what scenarios your method performing better?

---

### Official Review · Reviewer_wvUJ · 2025-11-02

**Soundness:** 2
**Presentation:** 2
**Contribution:** 3
**Rating:** 6
**Confidence:** 3

**Summary:**

This paper tries to tackle reward modeling by controlling the activated attention heads: learning a mask over them, running a PCA over the activations, and then scoring the reward by the likelihood of the model returning "Yes" to the question "Does this response meet the specified criteria". This method avoids having to train the model further or adapt the architecture. They show that this leads to improvements in reward modeling on reward bench as well as a new preference hacking benchmark they introduce.

**Strengths:**

This is an interesting paper that provides a new (to my knowledge) mechanism for defining reward models. Mod my concern about statistical significance, this is an interesting method that takes advantage of specialization amongst attention heads and appears to generate improved results.

**Weaknesses:**

# Serious issues / unresolved questions
- Which subset of attention heads are selected over during the training phase? All layers?
- My biggest concern is that this type of approach could increase the variance of the outcomes, but there are no std. errors indicating the variation across the dataset. This makes it hard to tell if the improvements are real.

# Minor issues
- The biggest writing issue I have reading this paper is that it has just moved many of the relevant technical details to the appendix. That's an authorial choice, and you don't have to fix it to improve my score, but I think section 3.2, which is a significant fraction of your method, should be more self-contained.
- Section B.2., motivating the emphasis on reward hacking, should almost certainly be in the main paper. As I was reading the paper, I was confused why reward hacking benchmarks were suddenly introduced as opposed to standard preference learning benchmarks?
- In table 3 I think "mean activations addition" is not a particularly clear term.
- In figure 1, there are two green boxes at the top. I assume that the masked out component is the activation reward model but it's not 100% clear. You might want to somehow make that clearer than just unifying them by color.
- Line 428-429 (and elsewhere that makes a similar claim) makes a stronger claim than I think you have evidence to support. It is the case that your method improves after you apply the PCA, but it's not obvious that it does so through the mechanism of reducing noise. Tbh, I'm not sure what other mechanism it might run through but I don't think one can make claims that don't have experiments to support them.
- I'm not sure I understand the lines "However, a clear advantage of our approach is the interpretability of using few-shot examples of a task to specify a reward signal. Thus, our approach is both a more aligned and interpretable reward score for model alignment." What is more interpretable about your methods?

**Questions:**

- In the appendix, for table 5, you have a claim that increasing the number of shots per sample leads to improved performance. What evidence is there for this? It seems like the improvement with increased number of shots is all over the place.
- Would it be fair to say, given table 3, that the main benefits are coming from the PCA filtering? Is that ablation done without the masking? It's a little unclear if each of the ablations stacks on top of the previous one or if they're all independent. For example, if just "yes" scoring was enough to get the boost, that'd be pretty concerning.

---

### Note · Authors · 2025-11-21

**Comment:**

Withdraw

**Withdrawal Confirmation:**

I have read and agree with the venue's withdrawal policy on behalf of myself and my co-authors.